


**Impact of glacial isostatic adjustment on zones of potential grounding line stability in the**
**Ross Sea Embayment (Antarctica) since the Last Glacial Maximum**
Samuel T. Kodama[1], Tamara Pico[1], Alexander A. Robel[2], John Erich Christian[3], Natalya Gomez[4],
Casey Vigilia[5], Evelyn Powell[6], Jessica Gagliardi[1], Slawek Tulaczyk[1], Terrence Blackburn[1]
[1]Earth and Planetary Science, University of California Santa Cruz, Santa Cruz, CA, USA
[2] School of Earth and Atmospheric Sciences, Georgia Institute of Technology, Atlanta, GA, USA
[3]Department of Geography, University of Oregon, Eugene, OR, USA
[4] Earth and Planetary Sciences, McGill University, Montréal, Québec, Canada
[5]Jackson School of Geosciences, University of Texas at Austin, Austin, TX, USA
[6] Lamont-Doherty Earth Observatory of Columbia University, Palisades, NY, USA
*Correspondence to*: Samuel T. Kodama (sakodama@ucsc.edu)
**Abstract**
Ice streams in the Ross Sea Embayment (West Antarctica) retreated up to 1,000 kilometers since
the Last Glacial Maximum, constituting one of the largest changes in deglacial Antarctic Ice
Sheet volume and extent. One way that bathymetry influenced this retreat was through the
presence of local bathymetric highs, or "pinning points", which decreased ice flux through the
grounding line and slowed grounding line retreat. During this time, glacial isostatic adjustment
vertically shifted the underlying bathymetry, altering the grounding line flux. Continental scale
modeling efforts have demonstrated the impact of solid Earth-ice sheet interactions on the
deglacial retreat of marine ice sheets, however, these models are too coarse to resolve small scale
bathymetric features. We pair a high-resolution bathymetry model with a simple model of
grounding line stability in an ensemble approach to predict zones of potential grounding line
stability in the Ross Sea Embayment for given combinations of surface mass balance rate, degree
of ice shelf buttressing, basal friction coefficient, and bathymetry (corrected for glacial isostatic
adjustment using three different ice sheet histories). We find that isostatic depression within the
interior of the Ross Sea Embayment during the Last Glacial Maximum restricts zones of
potential grounding line stability to near the edge of the continental shelf. Zones of potential
grounding line stability do not appear near the present-day grounding line until sufficient uplift
has occurred (mid-Holocene; ~5 ka), resulting in a net upstream migration of zones of potential



grounding line stability across the deglaciation. Additionally, our results show that coarse
resolution bathymetry underpredicts possible stable grounding line positions, particularly near
the present-day grounding line, highlighting the importance of bathymetric resolution in
capturing the impact of glacial isostatic adjustment on ice stream stability.

**1.  Introduction**
Since the Last Glacial Maximum (LGM) at approximately 26-19 ka (Clark et al., 2009), ice
streams in the Ross Sea Embayment sector of West Antarctica retreated up to 1,000 kilometers to
their present-day grounding line positions. Bathymetry can influence retreat of a marine ice sheet
by guiding grounding-lines through deep submarine troughs (Halberstadt et al., 2016; Jones et al.,
2021) or by slowing retreat through local bathymetric highs known as "pinning points". Pinning
points, either a product of antecedent bathymetry or sediment deposited in the form of grounding
zone wedges (Bart et al., 2018; Bart and Tulaczyk, 2020; Jamieson et al., 2012; Simkins et al.,
2018), can slow or pause ice sheet retreat by decreasing ice flux through the grounding line
(Jamieson et al., 2012; Robel et al., 2022; Schoof, 2007).
Over the last deglaciation, the bathymetry underlying the marine-based (grounded below sea
level) Ross Sea Embayment of the Antarctic Ice Sheet has been modulated by glacial isostatic
adjustment (GIA), the solid Earth's response to ice sheet unloading through crustal deformation
and perturbations to the Earth's gravitational field and rotation axis (Kendall et al., 2005). Changes
to bathymetry caused by these solid Earth-ice sheet interactions have been found to reduce the
modeled retreat rate for marine sectors of the Antarctic Ice Sheet, including over the last
deglaciation (de Boer et al., 2014; van Calcar et al., 2023; Gomez et al., 2013, 2015, 2018; Konrad
et al., 2015; Pollard et al., 2017). Although these coupled models provide valuable insight into ice
sheet-solid Earth interactions, they are computationally expensive with limited capacity to explore
a large parameter space while still resolving bathymetry geometry and the grounding line at high
resolution.
To better understand the impact of GIA on Antarctic deglaciation, it is necessary to explore
how GIA modulates both large- and small-scale bathymetric features. Here we use an ensemble of
simple grounding line stability calculations to take advantage of high-resolution bathymetry
models and isolate the impact of bathymetric change due to GIA on ice stream grounding line
stability during the deglaciation of the Ross Sea Embayment. We model grounding-line stability





along 147 LGM ice stream flowlines in the Ross Sea Embayment, over present-day and 20 ka
GIA-corrected bathymetry. Rather than reconstructing an exact history of Ross Sea Embayment
grounding-line evolution, we predict zones of potential grounding line stability (henceforth termed
"zones of potential stability"), at 20 ka and present-day. Improved information about ice margins
in the past, in conjunction with our predictions for zones of potential stability, could guide future
identification of locations where past Ross Sea Embayment ice stream grounding lines were likely
to persist. We explore the contribution of GIA to grounding line stability at present-day and 20 ka
grounding line locations, quantifying the impact of GIA on zones of potential stability across the
deglaciation.

**2.  Methods**
**2.1 Modeling Ross Sea Embayment Paleobathymetry**

To reconstruct Ross Sea Embayment 20 ka paleobathymetry we modify present-day

BedMachine bathymetry (500 m horizontal resolution; Morlighem et al., 2020) for the
spatiotemporal patterns of GIA caused by the deformational, gravitational, and rotational effects
associated with changes in ice load. Sedimentation would have also altered the paleobathymetry
of the Ross Sea Embayment since the LGM, however the magnitude of sedimentation across the
Ross Sea Embayment is still poorly constrained and so we focus on the role of GIA. Our
simulations are based on the sea-level theory and pseudo-spectral algorithm described by Kendall
et al. (2005), with a spherical harmonic truncation at degree and order 512 (spatial resolution of
~40 km). This treatment includes the impact of load-induced Earth rotation changes on sea level
(Milne and Mitrovica, 1996), evolving shorelines, and the migration of grounded, marine-based
ice (Johnston, 1993; Kendall et al., 2005; Lambeck et al., 2003; Milne et al., 1999). Our GIA
simulations require two inputs: (1) an Earth structure model with a depth-varying viscosity of the
mantle along with an elastic lithospheric thickness; and (2) the space-time geometry of ice sheet
thickness. The resulting GIA output varies smoothly across spatial scales much broader than the
spatial scales of Ross Sea Embayment bathymetric changes.

The LGM extent and deglacial history of Antarctica is uncertain due to a paucity of datasets

constraining past ice thickness and sea-level change (Clark and Tarasov, 2014; Deschamps et al.,
2012; Golledge et al., 2014; Gomez et al., 2018, 2020; Lambeck et al., 2014; Lin et al., 2021;
Pittard et al., 2022; Simms et al., 2019; Whitehouse et al., 2012). To represent these uncertainties,



we use three different ice-sheet histories that span a range of LGM ice-sheet thickness
reconstructions. The first ice history Golledge et al. (2014; henceforth Gol14) contains a deglacial
Antarctic Ice Sheet volume change of ~10.5 m global mean sea level equivalent (GMSLE) and
was created from the median of an ensemble of Parallel Ice Sheet Model runs (Bueler and Brown,
2009) forced by an Earth system model and uniform sea-level changes. The second ice history
Whitehouse et al., (2012; henceforth W12) contains a deglacial Antarctic Ice Sheet volume change
of ~8 meters GMSLE and was created by running the GLIMMER ice sheet model (Rutt et al.,
2009) for discrete time intervals (20, 15, 10, and 5 ka), and is constrained by glaciologic, geologic,
and GIA records. The third ice history Gomez et al. (2018; henceforth Gom18) has a deglacial
Antarctic Ice Sheet volume change of ~6 m GMSLE. The Gom18 model is a coupled,
gravitationally consistent GIA-dynamic ice sheet model that incorporates 3-D earth structure and
was forced by climate via benthic $\delta^{18}O$ records.
These ice-sheet histories encompass a range of potential Antarctic Ice Sheet volume
changes (6–10.5 m GMSLE; Supplementary Figure S1) and therefore, a range of potential GIA
magnitudes across the Ross Sea Embayment. To simulate GIA for W12 and Gol14 we use a 1-D,
radially symmetric Earth model with lithospheric thickness of 96 km, upper mantle viscosity of
$10^{21}$ Pa s and lower mantle viscosity of $10^{22}$ Pa s, similar to the best fit 1-D Earth model used in
Whitehouse et al. (2012). For Gom18 we use the 3-D Earth model GIA output from Gomez et al.
(2018). We explore the sensitivity to our choice in Earth model by simulating GIA for W12 and
Gol14 using the VM5a Earth model (Peltier et al., 2015) and a lower viscosity Earth model more
representative of West Antarctica, characterized by a 50 km lithosphere and low viscosity zone of
$10^{19}$ Pa·s from 50 km down to 200 km depth. For the Gom18 ice history we explore sensitivity to
Earth model by comparing to the 1-D reference Earth model (Gomez et al., 2018).

**2.2 Simulating Grounding line Stability**

To determine the locations of potential zones of grounding line stability in the Ross Sea
Embayment, we first trace 147 LGM ice stream flowlines based on reconstructions of Anderson
et al. (2014) and present-day ice-sheet flow (Rignot et al., 2011). We consider the 20 ka
paleobathymetry and present-day bathymetry to compare LGM and interglacial endmembers.
Although Ross Sea Embayment ice stream flowlines underwent reorganization throughout the last
deglaciation (Greenwood et al., 2018; Lee et al., 2017), we use LGM flowlines when simulating





grounding line stability for both present-day (isostatically rebounded) and 20 ka (isostatically
depressed) bathymetry to make clear comparisons between these time periods. By using LGM
flowlines for both present-day and 20 ka (GIA-corrected) bathymetry we can also test for potential
zones of grounding line stability throughout the entire Ross Sea. Although using LGM flowlines
for present-day can result in predicting zones of potential stability at locations that contradict the
geologic record (e.g. predicting zones of potential stability on present-day bathymetry offshore of
the present-day grounding line), it expands our understanding of solid Earth-ice sheet interactions
in a way that would not be possible with traditional ice-sheet modeling methods.
We simulate potential locations of grounding line stability for each ice stream flowline by
first extracting its bathymetric profile for present-day and 20 ka GIA-corrected bed bathymetries
with three ice histories. We then use a simple differential equation for the mass balance of a marine-
terminating ice stream (Equation 1 in Robel et al., 2018, following on Schoof 2012) to test for
stability along each ice stream flowline at 1 km-spaced nodes. We consider different combinations
of accumulation, basal friction, and ice-shelf buttressing parameters (Table 1); resulting in an
ensemble of 1,000 stability calculations at each node along an ice stream flowline. The model is
given by
$$h_g \frac{dL}{dt} = PL - \Omega h_g^\beta \tag{1}$$


in which
$$\beta = \frac{m + n + 3}{m + 1} \tag{2}$$

$$\Omega = \frac{A(\rho_i g)^{n+1} \left(1 - \frac{\rho_i}{\rho_w}\right)^n}{(4^n C)^{\frac{1}{m+1}}} \Theta^{\frac{n}{m+1}} \tag{3}$$

$$h_g = -b \frac{\rho_w}{\rho_i} \tag{4}$$

Where P is the upstream averaged surface mass balance $\left(0.01 - 0.3 \frac{m}{yr}\right)$, L is the distance of the
node downstream from the ice divide, t is time, $h_g$ is ice thickness at the grounding line, and $\Omega$ is
a scalar that accounts for factors that affect grounding line flux such as basal friction
($1.62 - 6.62\ Pa \cdot s^{-1}$), ice-shelf buttressing (0.5–1.0; smaller values representing more
buttressing). A is the Nye-Glenn law coefficient ($2x10^{-24}\ Pa \cdot s^{-1}$), n is the accompanying Nye-





Glen law exponent (3), m is the Weertman friction law exponent ($\frac{1}{3}$), $\rho_i$ is the density of ice (917
$\frac{kg}{m^3}$), $\rho_w$ is the density of sea water (1028 $\frac{kg}{m^3}$), and g is gravitational acceleration (9.81 $\frac{m}{s^2}$).

We allow for ice-shelf buttressing given evidence for the formation of ice shelf

embayments in both East and West Ross Sea (e.g. Bart et al., 2018; Prothro et al., 2020), and our
average upstream surface mass balance values are consistent with local ice core and ice penetrating
radar records (Buizert et al., 2015; Cavitte et al., 2018). The form of the grounding line flux ($\Omega h_g^\beta$)
is taken from prior asymptotic approximation studies of ice flux at grounding lines (e.g., Schoof,
2007; Haseloff and Sergienko, 2018) and is appropriate to use here since we only analyze the
steady-state of Equation 1. The form used here assumes a given Weertman-style basal friction law,
no lateral shear stress, and buttressing from an ice shelf of fixed size. While other forms of the
grounding line flux (Haseloff and Sergienko, 2018) may be used, the results here are unlikely to
be strongly dependent on the particular form used as long as there is a strong dependence on
bathymetry (as occurs in existing grounding line flux approximations, e.g. Schoof 2007).

A node is a potential "stable steady-state" if two conditions are met: (1) the ice flux into

the node is equal to ice flux out ($\frac{dL}{dt} = 0$ in Equation 1), and (2) the first derivative of the right-
hand side of Equation 1 with respect to L is negative. The latter condition means perturbations to
the grounding line position return the grounding line to its original position. This is expressed as:
$$Ph_g^{-1} + \left[PLh_g^{-2} + (\beta - 1)\Omega h_g^{\beta-1}\right]\frac{\rho_w}{\rho_i}\frac{dh}{dL} < 0 \tag{5}$$
These conditions constitute a linear stability analysis of the grounding line position (Schoof
2012; Robel et al. 2018).





| Table 1: Parameters used in Equations 1-5 | | |
|---|---|---|
| Parameter | Description | Value |
| L | Distance downstream from ice divide (m) | - |
| P | upstream average surface mass balance $(\frac{m}{yr})$ | 0.01-0.3 |
| $h_g$ | ice thickness at the grounding line (m) | - |
| h | Topographic elevation at the grounding line (m) | - |
| A | Nye-Glen law coefficient $(Pa^{-n} \cdot s^{-1})$ | $2x10^{-24}$ |
| m | Weertman friction law exponent | $\frac{1}{3}$ |
| n | Nye-Glen law exponent | 3 |
| C | Basal friction coeffiscient $(Pa \cdot m^{\frac{-1}{n}} \cdot s^{\frac{1}{n}})$ | 1.62 - 6.62 x $10^6$ |
| $\Theta$ | Ice shelf Buttressing parameter | 0.5-1 |
| $\rho_i$ | Ice density $(\frac{kg}{m^3})$ | 917 |
| $\rho_w$ | Sea water density $(\frac{kg}{m^3})$ | 1028 |
| g | Gravitational acceleration $(\frac{m}{s^2})$ | 9.81 |


**Table 1 | Parameters and values used for simulating grounding line stability.**

For each ice stream flowline bathymetry, our analysis of the ensemble predicts the locations

of potential stable grounding line positions for different combinations of surface mass balance (P),
ice shelf buttressing (θ), and basal friction coefficient (C; Table 1). Given the wide range of
uncertainty in climate and glaciological parameters for Antarctica during the LGM, our analysis
of a simple computationally efficient modeling approach allows us to sample a wide range of
parameter space that is unfeasible with more complex marine-terminating glacier models. The
zones of potential stability represent regions along the transect of bathymetry (either at present-
day or at 20 ka) where—for a combination of P, θ and C—an ice stream grounding line could
persist for an extended period of time, not necessarily where the grounding line is predicted to be
at any given time geologically.



As prior studies have argued (Robel et al., 2022; Sergienko and Haseloff, 2023), the
dynamic nature of climate, ice sheets, and the solid Earth make it unlikely that grounding lines
will achieve a mathematically stable steady-states in the real world. However, linear stability
analysis provides a useful guide to locations at which grounding lines are likely to persist or slow
down retreat for prolonged time periods. Our approach allows us to identify zones along a given
bathymetric transect where grounding lines may have persisted, without information about where
the ice margin existed geologically at any time, since information about the age and location of
past grounding lines is uncertain. Once we predict zones of potential stability along a bathymetric
transect, we explore the impact of grid resolution by resampling the bathymetric transect at
progressively coarser grid resolutions and repeating the stability analysis. We resample by
smoothing the transect to the desired coarse resolution and then resampling the smoothed
bathymetry.
To validate the results from our simple model of grounding line stability, we also modeled
transient grounding line evolution with a 1-D flowline model of marine ice-sheet evolution, using
the shallow-shelf approximation (Robel, 2021; Schoof, 2007). This model provides an alternative
method to identifying the role of GIA for transient cases that do not attain a strict steady-state, and
which do not require most of the assumptions intrinsic to Equation 5 (e.g. grounding line dynamics
are a smooth system). We calculate grounding line retreat rates and grounding line discharge over
present-day and 20 ka GIA-corrected paleo bathymetry and compare spatial patterns of transient
grounding line retreat and discharge between present-day and 20 ka GIA corrected bathymetries
(Supplementary Material). We find decreases in grounding line discharge and slowed grounding
line retreat rates at the same locations where we predict zones of potential stability with the simpler
model (Supplementary Material; Figure S2), which is not entirely surprising since the model
captured by Equations 1-3 is designed to approximate more complex ice sheet models at steady
states.
**3. Results and discussion**
**3.1 Patterns of GIA across the Ross Sea Embayment**
From 20 ka to present-day, all three ice histories result in relative sea-level (RSL) fall due
to GIA uplift of the Ross Sea Embayment interior (Figure 1a–c). The ice histories with larger
excess LGM ice volume cause larger RSL change, with a maximum RSL change of -140 m and -
170 m (Figure 1a; Figure 1b), for W12 (GMSLE = 8 m) and Gol14 (GMSLE = 10.5 m; Figure



S3), respectively. The interior of the Ross Sea Embayment, offshore of the Siple Coast (SC, Figure
1e) experiences the maximum uplift in both of these ice histories. In contrast, Gom18 produces a
smaller magnitude of uplift in the Ross Sea Embayment (115 m; Figure 1c), with two centers of
maximum uplift: one offshore of the Siple Coast (Figure 1c) and the other near Northern Victoria
Land (NVL; Figure 1c). The different pattern of GIA-induced uplift results from the lower
viscosity in the Ross Sea Embayment used in Gom18 compared to the higher-viscosity 1-D Earth
structure used in simulations of W12 and Gol14, in addition to differences in the ice loading history
(Figure 1a–c; Figure S4; Gomez et al., 2018).

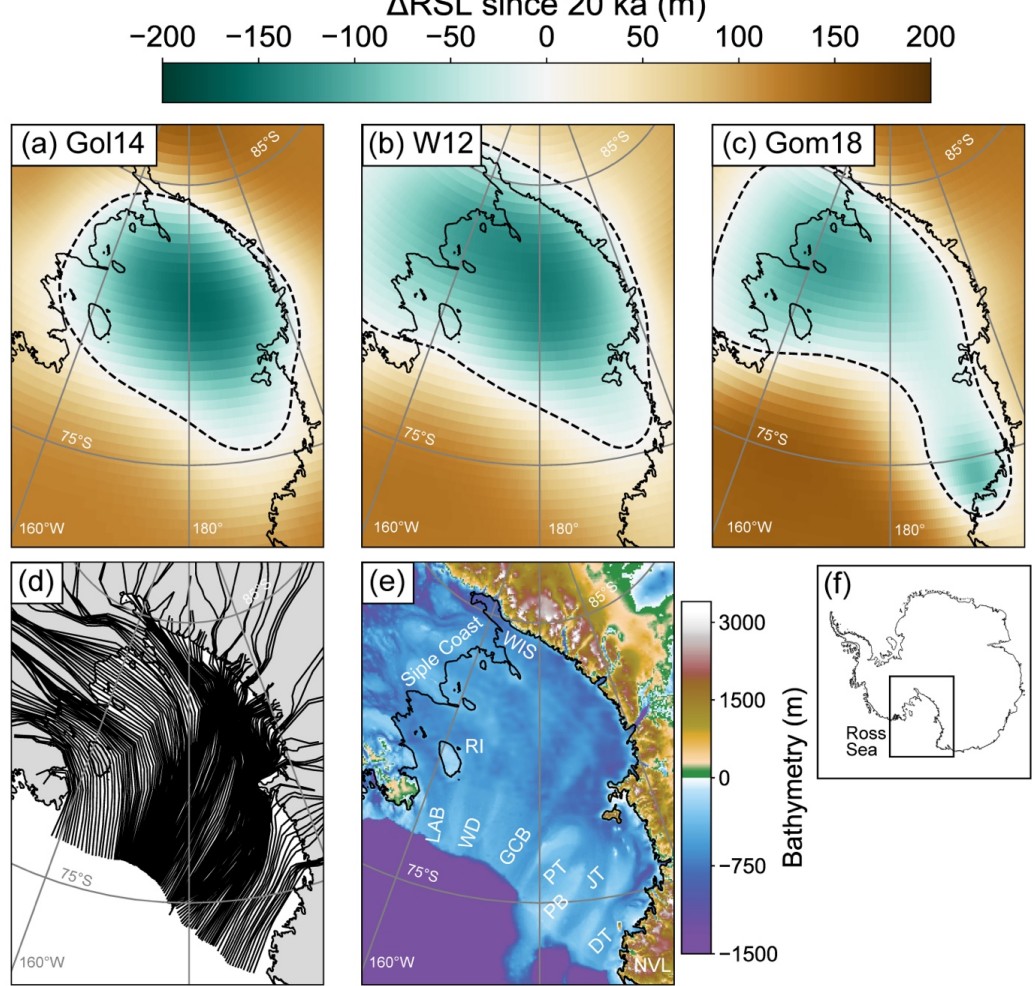






**Figure 1 | Change in relative sea level from 20 ka to present-day for a) Gol14 b) W12 and c) Gom18. Solid black line is the present-day grounding line and dashed black line marks the zero contour line of ΔRSL since 20 ka. d) Last Glacial Maximum ice stream flowlines based on Anderson et al (2014) and Rignot et al., (2011). e) BedMachine bed bathymetry and bathymetry of the Ross Sea Embayment (Morlighem et al., 2020). LAB – Little America Basin, WD – Wales Deep, GCB – Glomar Challenger Basin, PT – Pennell Trough, PB – Pennell Bank JT – JOIDES Trough, DT – Drygalski Trough, RI – Roosevelt Island, NVL – Northern Victoria Land, WIS – Whillans Ice Stream. f) Inset of Antarctica. Black box shows extent of panels a-e.**

**3.2 The impact of GIA on Grounding Line Stability from 20 ka to present-day**

Next, we quantify the impact of GIA on zones of potential stability. For each ice stream, our ensemble analysis predicts the locations of potential stable grounding line positions for a given bathymetry (20 ka and present-day) and set of parameter values. For example, Figure 2b shows the reconstructed bathymetric transect corrected for GIA (Gom18–orange, W12–purple, Gol14–red; Figure 2b) compared to the present-day bathymetry (black; Figure 2b) for the paleo-Whillans ice stream (Fig. 2a). Locations with more zones of potential stability are stable across a wider range of input parameter combinations, and therefore have a relatively higher likelihood of being stable regardless of parameter uncertainty. The present-day bathymetry has the majority of zones of potential stability located near the present-day grounding line (~750 km downstream; black; Figure 2b), whereas each 20-ka bathymetry has the majority of zones of potential stability near the continental shelf break (~1600 km downstream; orange, purple, and red; Figure 2b).

The shift of zones of potential stability across the deglaciation can be quantified by calculating the percent change in the number of potential stable grounding lines within a given reach of an ice stream transect from LGM to present-day. In Figure 2c-e we calculate the percent change of zones of potential stability along eight flowlines divided into 50 km reaches, spanning the Ross Sea Embayment from LGM to present-day, and find that zones of potential stability decrease near the continental shelf break, and increases further upstream, similar to predictions for the paleo-Whillans ice stream.

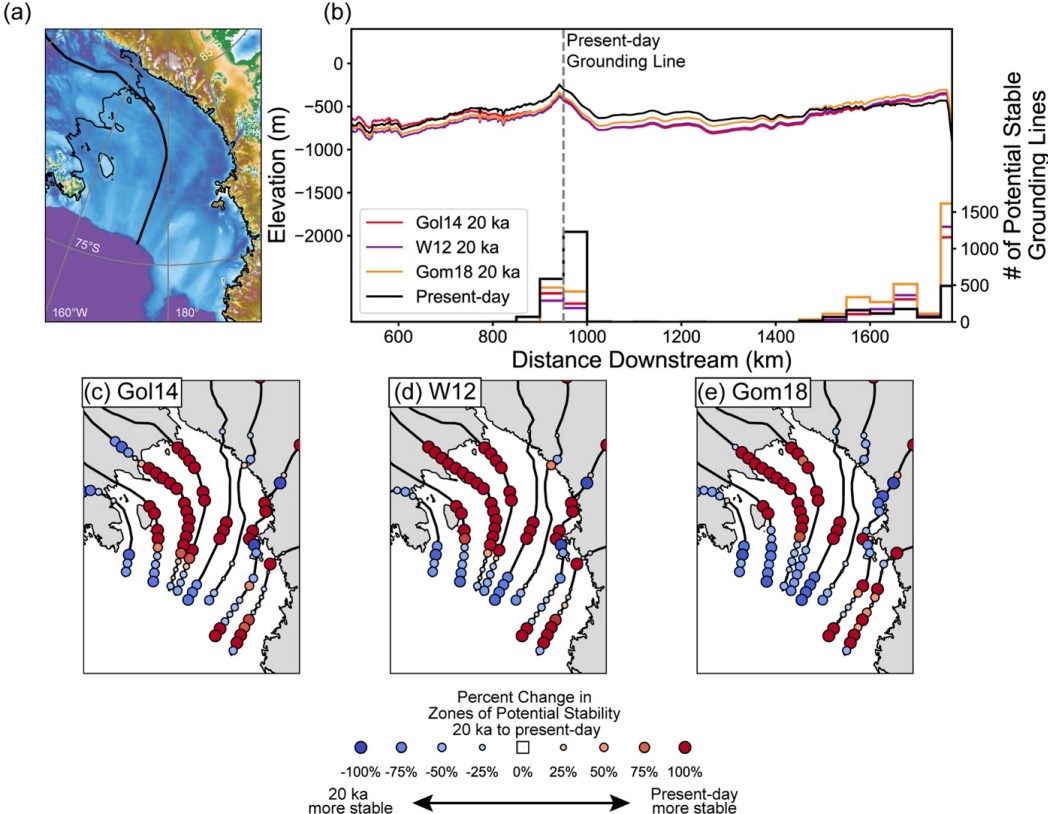

**Figure 2 | Changes in the density of potential zones of grounding line stability along a single flowline. a) Flowline path of the paleo-Whillans ice stream. b) Present-day bathymetry (black), and 20 ka GIA-corrected paleobathymetry for W12 (purple), Gol14 (red), and Gom18 (orange) along the paleo-Whillans ice stream flowline with corresponding histograms of simulated stable grounding line positions. Histograms and corresponding bathymetry share color. c-e) Percent change in the number of modeled potential stable grounding line from 20 ka (LGM) bathymetry to present-day bathymetry for Gol14 (c), W12 (d), and Gom18 (e) ice histories.**

We then expand our potential stable grounding line analysis to all 147 ice stream flowlines. We bin potential stable grounding line zones into a 20 km x 20 km grid to create fields of zones of potential stability for present-day and 20 ka paleo-bathymetry and calculate the percent change of zones of potential stability from 20 ka to present-day (Figure 3). We generally find that, from 20 ka to present-day, there are less zones of potential stability near the edge of the continental shelf (these locations were more stable during the LGM; blue; Figure 3), and more zones of potential stability upstream, signifying these locations are more stable during present-day (red and cyan;





Figure 3). The pattern of offshore decrease in zones of potential stability is interspersed with
isolated regions of increased or minimal change in zones of potential stability, corresponding to
ridges separating the Little America Basin, Wales Deep, and Glomar Challenger Basin (LAB, WD,
GCB; Figure 4c; Figure 1e). The Gom18 model predicts slightly different patterns compared to
the Gol14 and W12 models, with the decrease in zones of potential stability extending further
upstream and a larger area of zones of potential stability increase near the JOIDES and Drygalski
troughs (JT, DT; Figure 1e) caused by the second center of maximum uplift.

There are two GIA mechanisms that cause the locations of zones of potential stability to

shift upstream over the last deglaciation: 1) sea-level fall caused by rebound of the solid Earth
under the locus of ice mass loss and 2) sea-level rise caused by far-field ice sheet melt (and
secondarily by the collapse of the Antarctica peripheral bulge). Isostatic rebound within the interior
of the Ross Sea Embayment shoals bathymetry, decreasing flux through the grounding line, thus
increasing the potential for a "steady-state" grounding line. Sea-level rise caused by far field ice
sheet melt and Antarctic peripheral bulge collapse causes increased grounding line flux, which
decreases the likelihood of a "steady state" grounding line position near the edge of the continental
shelf (Figure 3).
Together, these GIA mechanisms cause a net upstream migration of zones of potential stability;
however, this predicted upstream migration differs from GIA forcing a transient grounding line
retreat throughout the deglaciation, and instead provides a window into how GIA stabilizes the
grounding line at each time frame. During the LGM the grounding line was located near the edge
of the continental shelf, and at present-day the grounding line is located within the Ross Sea
Embayment interior.  Our analysis shows that, when forced by GIA alone, the edge of the
continental shelf was more stable during the LGM (20 ka), and geologic records show that during
the LGM, the Ross Sea Embayment grounding line was located near the edge of the continental
shelf (Halberstadt et al., 2016; Prothro et al., 2020). Our analysis also shows that the interior of the
Ross Sea Embayment is more stable at present-day near locations of the present-day grounding
line. The co-occurrence of where GIA promotes stability within the Ross Sea Embayment with the
inferred location of the grounding line for both present-day and LGM, demonstrates that GIA
provides stability for the grounding line at both glacial maximum and interglacial climate states.

Near complete loss of zones of potential stability from 20 ka to present-day within the deep

submarine troughs of the Ross Sea Embayment suggest that far-field Northern Hemisphere ice-



sheet growth (and resulting global sea-level fall) is an important factor for stabilizing the LGM
grounding line at the continental shelf, therefore permitting larger LGM ice volume in the Ross
Sea Embayment (Supplementary Figure S3). Our finding, that zones of potential stability increase
at the Ross Sea Embayment LGM grounding line due to GIA, parallels the finding that far-field
ice-sheet retreat has an important feedback on Antarctic Ice Sheet deglaciation (Gomez et al.,

2020).

At present-day, GIA has caused the interior of the Ross Sea Embayment to rebound,

resulting in the emergence of zones of potential grounding line stability co-located with the
present-day grounding line. These zones of potential stability located near the present-day
grounding line are less prevalent at 20 ka due to isostatic depression, and predominately do not
emerge until the mid-Holocene (~5 ka; Supplementary Figure S4) for all ice histories, suggesting
that large magnitudes of isostatic rebound over a prolonged period (i.e., deglacial timescale)
provides stability at the present-day grounding line location.







**Figure 3 | GIA-induced percent change in stable grounding line positions across 20 km x 20 km grid cells for entire Ross Sea Embayment based on glacial isostatic adjustment simulations for a) Gol14, b) W12, and c) Gom18. Grid cells that have stable grounding line positions in the present-day and no stable grounding line positions at 20 ka are marked in teal. Thin black line is present-day day grounding line and dashed line is transition from sea-level fall (within) to sea-level rise (outside). d) Bathymetry of Ross Sea Embayment (Morlighem et al., 2020).**

**3.3 Characteristic Stable Grounding Line depth for the Ross Sea Embayment**

GIA moves zones of potential stability by modulating the bathymetry of the Ross Sea Embayment. However, bathymetry is only one of the parameters that determines where grounding line stability occurs. Grounding line stability is a function of surface mass balance (P), ice shelf buttressing (Θ), basal friction (C), distance downstream from the ice divide (L), and ice thickness at the grounding line ($h_g$). Therefore, for a given combination of input parameters (Table 1), there is an ice thickness $h_g$ (and corresponding grounding line depth) that produces a stable grounding line position (Figure 4a and Figure S5). Our grounding line stability analysis shows that stable grounding line depth is a function of the other input parameters (Figure 4a; Supplementary Figure S5). For instance, distance downstream from the ice divide and average upstream surface mass balance rate are both negatively correlated with grounding line depth (Figure 4a). Therefore, the depths at which we predict stable grounding lines are a function of both the deglacial climate system, and the geometry of the Antarctic Ice Sheet. As a result, the mean stable grounding line depths for LGM paleo-bathymetry and present-day bathymetry are similar (Figure 4a). These mean depths are significantly different from the bathymetry we input into the simple model of grounding line stability (Figure 4, grey; T-test; $p_{present-day}$, $p_{W12}$, $p_{Gol14}$, $p_{Gom18}$ < 0.001), showing that the depth range at which our simple model finds stability depends on our input parameters and the ice geometry of the Ross Sea Embayment. Therefore, we infer that zones of potential stability migrate spatially due to GIA by causing bathymetry to be uplifted or subsided into and out of this characteristic depth range (Figure 4c).





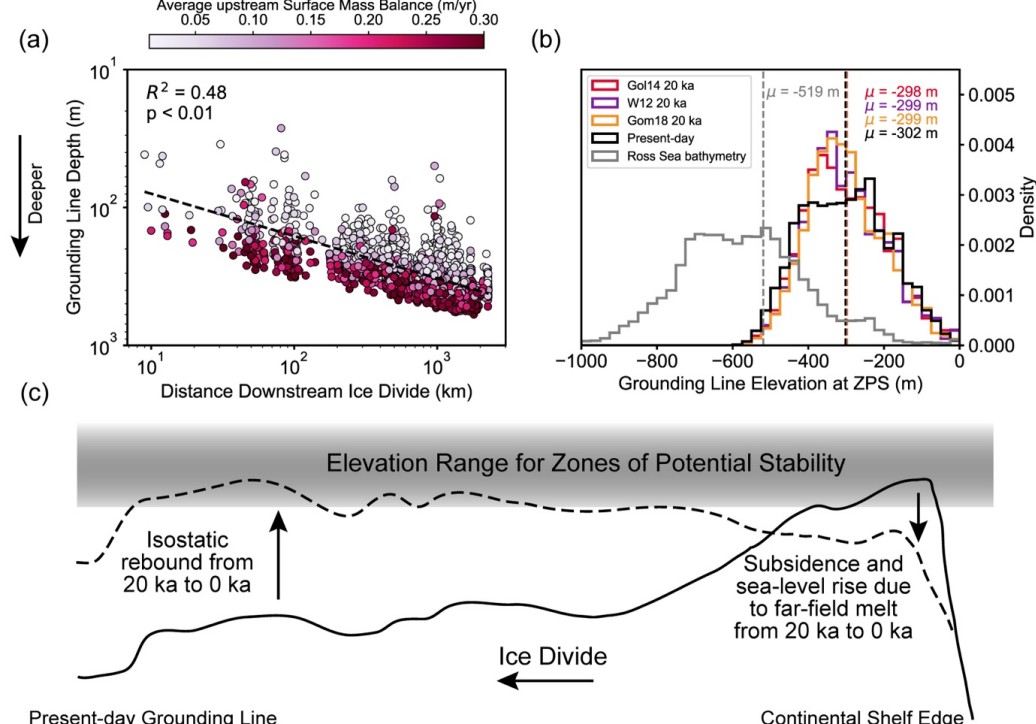

**Figure 4 | a) Average upstream surface mass balance at distance downstream of the ice divide and grounding line elevation for stable grounding line positions on present-day bathymetry. b) Histograms of elevations for the present-day Ross Sea Embayment bathymetry (grey), stable grounding line positions simulated over present-day bathymetry (black), and 20 ka paleobathymetry corrected for glacial isostatic adjustment based on the ice histories of Gol14 (red), Gom18 (orange), and W12 (purple). c) Schematic illustrating how glacial isostatic adjustment moves bathymetry into and out of potential stable grounding line elevations.**

**3.4 Influence of grid resolution on predicted zones of potential stability**

Coupled ice sheet-GIA models often use grid resolutions of 20-40 km (van Calcar et al., 2023; Gomez et al., 2018, 2020; Lowry et al., 2024) to reduce computational costs, however these grid resolutions do not resolve smaller-scale bathymetric features. To explore the effects of grid resolution, we use high resolution (500 m; Morlighem et al., 2020) bathymetry and resample to coarser resolution (20 km), comparing how predicted zones of potential stability vary across the Ross Sea (Fig. 5). Some of the datasets constraining bathymetry in the Ross Sea Embayment are gravity-based, and therefore have a true resolution coarser than the 500 m output resolution of BedMachine. Nonetheless, we decide to treat BedMachine as a 500 m resolution in the Ross Sea Embayment to explore the potential impacts of grid resolution. We predict fewer zones of potential





stability across the Ross Sea Embayment at LGM and present-day when using the coarse grid
resolution (20 km; Fig. 5b/c). Predictions with coarse resolution bathymetry largely fail to produce
zones of potential stability near the edge of the continental shelf (Fig. 5c). Furthermore, the coarse
resolution prediction fails to predict zones of potential stability near the present-day grounding
line on the present-day bathymetry (Fig. 5f).

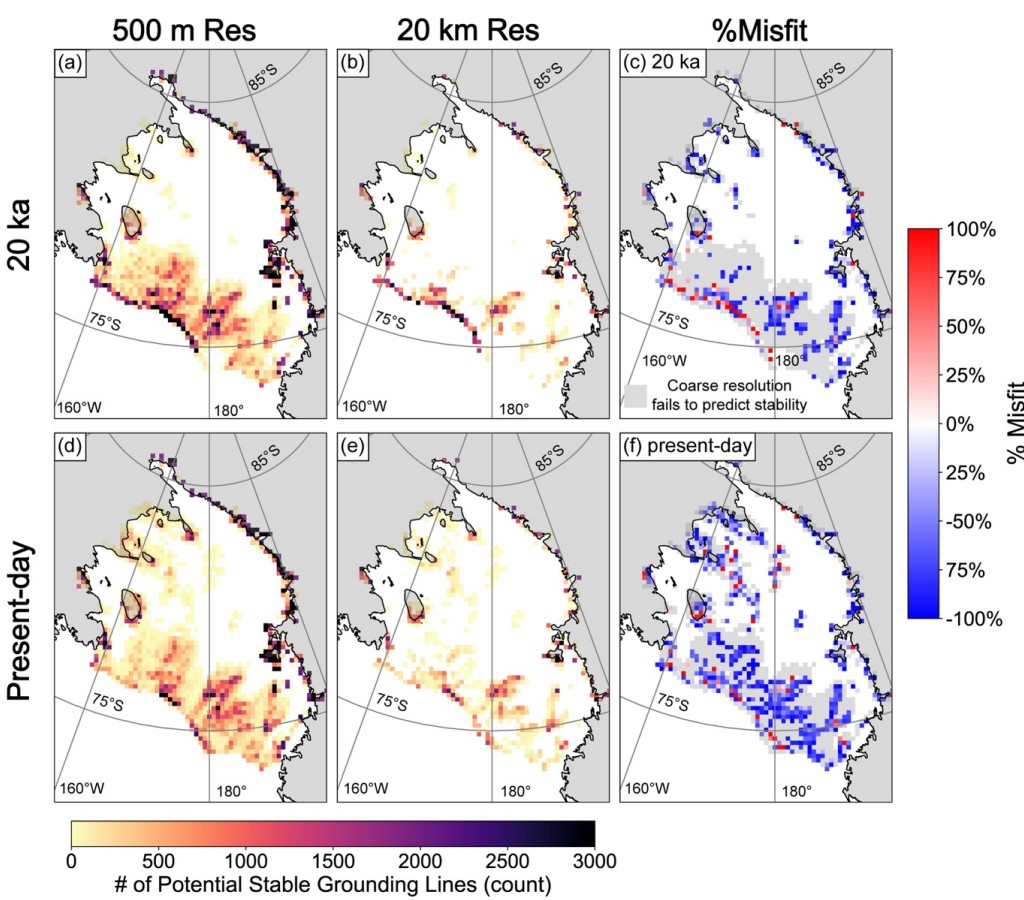


**Figure 5 | Density of stable grounding line positions for Gom18 Last Glacial Maximum (20**
**ka) (top row), Present-day (bottom row) for high resolution (50m; a/d) and low resolution**
**(20 km; b/e) Percent misfit for 20 ka (c) and present-day (f), defined as difference of stable**
**grounding line positions calculated using high resolution and coarse resolution bathymetry**
**divided by stable grounding line positions calculated using high resolution bathymetry**
$\left(\frac{ZPS_{500m}-ZPS_{20km}}{ZPS_{500m}}\right).$





377    Since coarse grid resolution leads to an underprediction of zones of potential stability for

378  LGM and present-day bathymetry, coarse grid resolution will also underpredict how GIA migrates

379  zones of potential stability across the deglaciation. We quantify how coarse grid resolution

380  underpredicts the impact of GIA by calculating the change in zones of potential stability from

381  LGM to present-day (Fig. 3), but with coarser grid bathymetric resolution (20 km). Figure 6a

382  shows the percent misfit for change in zones of potential stability of 20 km resolution compared

383  to 500 m resolution. We find that the coarse resolution bathymetry underpredicts the impacts of

384  GIA near the present-day grounding line (blue; Fig 6a) and fails to capture the impact of GIA

385  within most of the deep submarine troughs (grey; Fig 6a), suggesting that high resolution

386  bathymetry may be necessary to fully recognize the role of GIA in potential retreat and readvance

387  scenarios in the Ross Sea Embayment (Balco et al., 2023; Kingslake et al., 2018; Lowry et al.,

388  2024; Neuhaus et al., 2021; Venturelli et al., 2020). We also vary the coarseness of the grid

389  resolution and find a logarithmic relationship between bed resolution and percent misfit (compared

390  to 500 m bed resolution), highlighting the fact that understanding GIA and bathymetry interactions

391  requires a fine grid resolution. The importance of bed resolution is, in part, due to the broad scale

392  slope of the Ross Sea Embayment bathymetry, which is retrograde, and therefore a fine grid

393  resolution is needed to resolve small-scale bathymetric pinning points, which provide shallower

394  bathymetry that is more likely to be stable due to its shallow depth in addition to providing local

395  prograde slopes that are required to meet the second stability condition (Equation 5). Based on our

396  misfit analysis, resolving these pinning points requires a resolution on the kilometer to sub-

397  kilometer length scale (Fig. 6b).





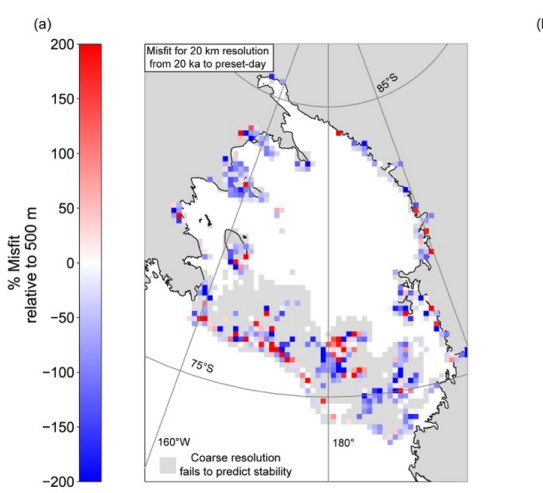
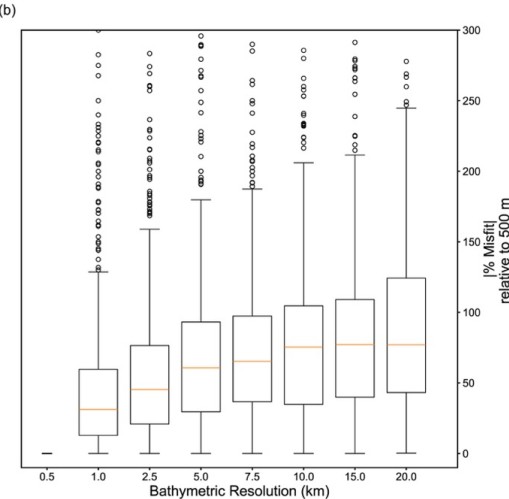

**Figure 6 | a) Quantification of how 20 km resolution bathymetric resolution underpredicts grounding line stability from 20 ka to present-day. Percent misfit defined as difference of stable grounding line positions calculated using high resolution and coarse resolution bathymetry divided by stable grounding line positions calculated using high resolution bathymetry $\left(\frac{ZPS_{500m}-ZPS_{20km}}{ZPS_{500m}}\right)$. b) Magnitude of percent misfit between zones of potential stability calculated with 500 m resolution bathymetry and coarser bathymetry resolutions.**

### 3.5 Comparisons with the Geologic Record of Grounding Line Retreat

The Ross Sea Embayment geologic record suggests a complex pattern of asynchronous retreat over the last deglaciation. In the western Ross Sea Embayment, retreat began in the Pennell Trough (PT; Fig. 1e) at ~15 ka and in the JOIDES Trough (JT; Fig. 1e) at ~13 ka (Prothro et al., 2020). Meanwhile, in the eastern Ross Sea Embayment, small-scale retreat in Wales Deep (WD Fig. 1e) also began at ~15 ka and increased during the early Holocene (Bart et al., 2018). Across the west and east Ross Sea Embayment, the ice sheet remained grounded to the trough banks, as the ice sheet retreated through submarine troughs, forming embayments (Halberstadt et al., 2016). The grounding line then retreated throughout the Holocene (Bart et al., 2018; Halberstadt et al., 2016; Prothro et al., 2020). Prior to ~8.6 ka, ice streams offshore Northern Victoria Land underwent reorganization (Greenwood et al., 2018; Lee et al., 2017). It is possible that local GIA uplift played a role in this reorganization (Figure 1c) as ~60 m of uplift occurred in this region from the LGM to the early Holocene (Figure S6).



A common pattern across ice sheet histories is a decrease in zones of potential stability within
the deep submarine troughs near the edge of the continent shelf (Figure 3), which occurs due to
sea-level rise driven by far-field ice-sheet melt. The ridges separating the submarine troughs are
shallow enough to stabilize the grounding line, despite relative sea-level rise during the
deglaciation (Figure 4c), which prevents a decrease in zones of potential stability along these
ridges. In the geologic record geomorphic features suggests that the grounding line back-steps up
these banks throughout the deglaciation (Halberstadt et al., 2016). Part of this back-stepping may
be caused by far-field sea-level rise, forcing the grounding line to backstep up the bank to shallower
depths, in addition to other drivers of retreat, such as ocean forcing.
**4. Conclusion**
Over the deglaciation, the Ross Sea Embayment experienced sea-level fall within its interior due
to glacial isostatic rebound, and sea-level rise near the edge of the continental shelf due to far field
sea-level rise, and secondarily due to peripheral bulge collapse. We use a simple model of
grounding line stability to show that glacial isostatic adjustment promotes grounding line stability
near the edge of the continental shelf at the Last Glacial Maximum and near the present-day
grounding line at the present-day, resulting in a net upstream migration of zones of potential
stability across the deglaciation. We also show that coarse bathymetric resolution causes an
underprediction of grounding line stability near the present-day grounding line and within the deep
submarine troughs of the Ross Sea, and thereby underpredicts the impact of glacial isostatic
adjustment on grounding line stability at these locations. This finding highlights the importance of
bathymetric resolution when modeling deglacial and potential grounding line re-advance scenarios
in the Holocene. Given the potential importance of small-scale bathymetric features in grounding
line stability within the Ross Sea Embayment, future work coupling a high-resolution regional ice
sheet model with a glacial isostatic adjustment model may provide insight into the role of glacial
isostatic adjustment and bathymetry in the Holocene retreat and readvance of the West Antarctic
Ice Sheet in the Ross Sea Embayment.  In this study we have shown that, in addition to influencing
transient grounding line retreat, glacial isostatic adjustment can promote stability across longer ice
age timescales by shifting zones of potential stability between the Last Glacial Maximum and
present-day ice sheet grounding zones. Our results highlight the role of topography, as modulated
by solid Earth processes, in shaping the history of ice sheet advance and retreat on glacial-
interglacial timescales.




**5. Author contribution**

Conceptualization – STK, TP, JG, ST, EP

Investigation – STK, TP, AAR, JEC, NG, CV

Formal Analysis – STK, JEC

Methodology– STK, TP, AAR, JEC, NG

Project Administration – TP, TB

Writing (original draft) – STK

Writing (review & editing) – STK, TP, AAR, NG, JEC, CV, EP, JG, ST, TB

**6. Acknowledgements**

We thank Pippa Whitehouse and Nicholas Golledge for sharing their ice sheet model output.

**7. Competing interests**

The authors declare that they have no conflict of interest.

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
