# Peer review of "Impact of glacial isostatic adjustment on zones of potential grounding line persistence in the"

_EGUsphere, 2024_

## Author Comment (AC1)

We thank the Editor and the Referees for their time and their positive and constructive feedback of the manuscript. In response to these reviews, we have edited the text to clarify our arguments and the results of the study. In particular, we have added text to the introduction that contextualizes the motivation of our study for a broader audience. Further, we have addressed all minor revisions suggested by the Referees. We believe that the revised manuscript is improved relative to the original.

In the following, we address each of the comments raised in the reviews and provide a detailed listing of the associated revisions to the text. We intersperse the reviewers' comments (black font) with our responses (blue font).

The authors investigate how glacial isostatic adjustment (GIA) has influenced zones of potential grounding line stability in the Ross Sea Embayment since the Last Glacial Maximum (LGM). They use a high-resolution bathymetry model combined with a simple grounding line stability framework to assess how GIA-induced changes in bathymetry affect ice sheet retreat and stabilization. Their analysis incorporates three different ice sheet histories and several Earth models to account for uncertainties in past ice loading and mantle viscosity.

The results show that during the LGM, grounding line stability was concentrated near the continental shelf edge, but as isostatic rebound progressed, stability zones migrated upstream, aligning with the present-day grounding line. They also find that coarse-resolution bathymetry underestimates grounding line stability, emphasizing the importance of high-resolution models. The authors conclude that GIA plays a crucial role in stabilizing ice sheets over long timescales and that bathymetric resolution significantly impacts the accuracy of grounding line predictions.

General remarks

This study presents an insightful analysis of the role of glacial isostatic adjustment (GIA) in grounding line stability within the Ross Sea Embayment. The authors effectively apply a high-resolution bathymetry model and an ensemble of simple grounding line stability calculations to assess how GIA-induced bathymetric changes influence grounding line migration. The results contribute valuable knowledge on the spatial and temporal evolution of stability zones, and the study convincingly demonstrates the importance of bathymetric resolution in capturing these effects. The paper is well-structured, with clear research objectives and a logical progression of results.
We thank the referee for this positive appraisal.

However, the introduction is quite brief and lacks sufficient discussion on how this study builds upon and differentiates itself from prior research on GIA and ice sheet stability. Providing more context on the relevance of grounding line stability zones and linking this work more explicitly to previous studies would strengthen the framing.

Thank you for this suggestion. We have added additional discussion within the introduction to provide background and frame the motivation of our study, which we highlight in the line-by-line comments below.

The methods section, while detailed, could benefit from additional clarification in several places, particularly regarding the ice sheet histories, Earth models, and parameter choices. Please see the line-by-line comments for the details.

We now provide more detail within our methods section, which is addressed in the line-by-line comments below.

Line-by-line comments

L53-58: The connection between the paragraphs can be more informative. Lines 53-56 explain that the retreat have been found to be reduced due to GIA and that the models cannot explore a large parameter space at high resolution, but it is not stated explicitly why one needs to explore the full parameter space and which remaining research questions are still open.

This comment is addressed with the comment for L59-61 below.

L59-61: Place your research in context here. Discuss other research that have been conducted in Antarctica to study how GIA modulates bathymetric features. Why do you not need coupled ice sheet – GIA to study the effect of bathymetry on grounding line migration? What is the method to obtain your ensemble of simple grounding line stability calculations? Have this method been applied in literature, maybe to other regions? Have the ensemble of simple grounding line stability calculations been used in other studies? Are there other regions, or other studies for the Ross Sea Embayment, where the effect of GIA on bathymetry have been studied using high resolution models?

The general feedback raised in these two comments are now addressed in an additional paragraph added to the introduction.

We add text explaining the benefit of ensembles (L 59-61):

"As a result, ensemble runs of ice sheet or coupled GIA-ice sheet models that can encompass uncertainties in climate and glaciologic conditions are run at relatively coarse resolution …"

We specify the open research question we are addressing (L 62-70):

"…and are unable to resolve smaller scale bathymetric features, such as pinning points (≤5 km; e.g., McKenzie et al., 2023), that may influence grounding line evolution. …Therefore, exploring the impact of high-resolution bathymetry on grounding evolution in the Ross Sea Embayment since the LGM is still a computational challenge."

We provide broader context regarding coupled GIA-ice sheet models to highlight the benefits of using a simpler modeling approach that allows for high resolution bathymetry (L 58-64).

"Although coupled GIA-ice sheet models provide valuable insight into ice sheet-solid Earth interactions across Antarctica, they are computationally expensive. As a result, ensemble runs of ice sheet or coupled GIA-ice sheet models that can encompass uncertainties in climate and glaciologic conditions are run at relatively coarse resolution (16-40 km; Albrecht et al., 2020, 2024; van Calcar et al., 2023; Gomez et al., 2018; Pollard et al., 2017), and are unable to resolve smaller scale

bathymetric features, such as pinning points (≤5 km; e.g., McKenzie et al., 2023), which influences grounding line evolution.

And note recent high resolution, coupled GIA-ice sheet models in other regions of Antarctica (L 64-68):

"Recent coupled GIA-ice sheet modeling showed that increasing bathymetric resolution (from 2 km to 1 km) slowed predictions of grounding line retreat by up to ~20% in the Amundsen Sea (Houriez et al., 2025); however such high-resolution modeling is more appropriate for smaller regions (the Ross Sea Embayment is ~4x their model domain size) and for timescales of centuries…"

L58: How large is the ensemble?
We now specify (L 71): "Here we use an ensemble (n=$9 \times 10^5$)"

L63-65: Explain why you made the choice to predict zones of potential grounding line stability and not reconstruct the exact history.
Our choice to model zones of potential grounding line persistence, rather than the exact history of grounding evolution, is driven by the opportunity to (1) sample a wide parameter space glaciologic conditions, and (2) understand the role of glacial isostatic adjustment on grounding stability across a broad topographic region (an entire flowline across the Ross Sea) regardless of the exact configuration of the past ice sheet. It is challenging to reconstruct the history of grounding line evolution over the last deglaciation in the Ross Sea; indeed, much of this history is unknown, leading to uncertainty in ice sheet reconstructions and associated glacial isostatic adjustment predictions. Our approach allows us to focus on a subset of the many controls on deglaciation, namely assessing the stability of possible grounding line locations across the entire Ross Sea for a range of glacial isostatic adjustment scenarios. The advantage of this approach is that we can identify multiple possible persistent grounding line locations, rather than being tied to a single specific grounding line evolution history.

We now explain (L 75-81):

"Rather than reconstructing an exact history of grounding-line evolution, we predict zones where a potential grounding line could persist at 20 ka and present day. This choice allows us to produce a probability distribution of locations where past ice stream grounding lines were likely to persist, which we term "zones of potential persistence", across the entire Ross Sea Embayment. We explore the contribution of GIA to grounding line persistence at present-day and 20 ka grounding line locations, quantifying the impact of GIA on zones of potential persistence across the deglaciation."

We also note (L 202-206):

"However, linear stability analysis provides a useful guide to identify locations where grounding lines were likely to have persisted or slowed down retreat for prolonged time periods without information about where the ice margin existed geologically at any time, since information about the age and location of past grounding lines is uncertain."

We also want to highlight a broad terminology change in the manuscript. In the original submission we included text to make distinctions between mathematical steady states and how our analysis provides locations at which grounding lines were likely to have persisted or slow down retreat for prolonged time periods. To further make this distinction clear we now use the term "zones of potential persistence" instead of "zones of potential stability".

L65-68: Explain in more detail the relevance of the identification of locations.
Identifying these locations, and how they change across the deglaciation provides examples of locations within the Ross Sea that might be of interest for future modeling efforts such as coupled models with mesh grids that can achieve locally high resolution at these areas of interest. For locations where drill campaigns have already occurred identifying these locations provides more context to view the geologic records from the drill campaigns.

L68: In the introduction, there is no mention of the contribution of GIA to grounding line stability at present-day. Please include a section in the introduction about ongoing GIA in the Ross Sea Embayment, how is it measured, how strong is the signal.
Since our study focuses on the GIA signal across the deglaciation and not the ongoing GIA signal, we choose to not discuss present-day observations, to limit potential confusion.

L75: Which version of Bedmachine?
We now specify in the main text (L 84-85).
> "To reconstruct Ross Sea Embayment 20 ka paleobathymetry we modify present-day BedMachine v1.38 bathymetry (500 m horizontal resolution; Morlighem et al., 2020)"

L94-104: Please provide more information on the ice thickness histories. How have these three histories been selected? Is the output of the Gol14 and Gom18 models constrained by observations? For example, how well does bedrock change at present-day from Gom18 match with GPS observations? What is the uncertainty of the benthic d18O records? Furthermore, why do the models vary in Antarctic ice sheet volume change? What is the spatial and temporal resolution of the output of all three models? How do the ice sheet histories exactly differ from each other and is one more realistic than the other? Why have W12 and Gom18 a sharp transition in ice thickness upstream and downstream of the grounding line, respectively. Might this effect your results?
Instead of providing a detailed review of each ice history, we refer the readers to the original papers, however we do now provide more information (**underlined in bold**) within our summaries of each model to contextualize the choice of ice histories (L 103-118)

> "To represent these uncertainties, we use three different ice-sheet histories that span a plausible range of LGM ice-sheet thickness reconstructions. The first ice history Golledge et al. (2014; henceforth Gol14) contains a deglacial Antarctic Ice Sheet volume change of ~10.5 m global mean sea level equivalent (GMSLE), **and was run at 14 km resolution and 100 year timesteps**. Gol14 was created from the median of an ensemble of Parallel Ice Sheet Model runs (Bueler and Brown, 2009) forced by an Earth system model and uniform sea-level changes, **and constrained**

**by geologic observations such as ice-core derived changes in regional ice thickness**. The second ice history Whitehouse et al., (2012; henceforth W12) contains a deglacial Antarctic Ice Sheet volume change of ~8 meters GMSLE and was created by running the GLIMMER ice sheet model (Rutt et al., 2009) for discrete time intervals (20, 15, 10, and 5 ka) **with a 20 km resolution, and is constrained by glaciologic, geologic, and Antarctic relative sea-level records**. The third ice history Gomez et al. (2018; henceforth Gom18) has a deglacial Antarctic Ice Sheet volume change of ~6 m GMSLE. The Gom18 model **is a single model run** of a coupled, gravitationally consistent GIA-dynamic ice sheet model that incorporates 3-D earth structure and was forced by climate via benthic $\delta^{18}O$ records. **The ice sheet model was run with 20 km resolution and 200 year timesteps.**"

Regarding the sharp transition in ice thickness upstream and downstream of the grounding line for W12 and Gom18, we do not know the exact origin of the difference, although it is likely due to differences in modeling approach and attempts to reproduce present-day conditions. However, given the long-wavelength signal of GIA, and given that the W12 and Gom18 output produce similar patterns of zones of potential stability offshore the Siple Coast (main text Figure 3), it is unlikely that this ice thickness transition impacts our analysis.

L109-110: Explain briefly how Whitehouse et al. (2012) determined the best fit 1D model.
We now explain (L 123-125):
> "…similar to the best fit 1-D Earth model used in Whitehouse et al. (2012), which was determined by inverting for the solid Earth rheology that best fit Antarctic deglacial sea-level data"

L111-114: To improve readability, move this sentence to line 110 to discuss it right after you mention the 1D model. Also, explain the VM5a Earth model and include a reference for the representative Earth model for West Antarctica.
We move the sentence and now include the reference for the representative Earth model (Pollard et al., 2017) and include the lithosphere thickness and upper and lower mantle viscosity values for VM5a (L 127-129).
> "VM5a Earth model which has a lithosphere thickness of 96 km, average upper mantle viscosity of 0.5 x $10^{21}$ and average lower mantle viscosity of 1.6 x $10^{21}$ (Peltier et al., 2015)"

> Pollard, David, Natalya Gomez, and Robert M. Deconto. "Variations of the Antarctic Ice Sheet in a Coupled Ice Sheet-Earth-Sea Level Model: Sensitivity to Viscoelastic Earth Properties." *Journal of Geophysical Research: Earth Surface* 122, no. 11 (2017): 2124–38. https://doi.org/10.1002/2017JF004371.

L114-115: The sensitivity analysis compares the 3D model to a 1D model, but a more comprehensive exploration of uncertainty would require varying the 3D Earth model itself, as it inherently contains uncertainties. Since mantle viscosity and lithospheric thickness cannot be measured directly, the 3D model is subject to assumptions and potential biases. Currently, the discussion does not acknowledge the uncertainties within the 3D model, which may affect the

results. It would be valuable to include a discussion on this limitation and its potential impact on the findings.

Instead of exploring uncertainty within the 3-D Earth model of Gom18, which arises from choice of lithospheric thickness as well as the method used to map seismic velocity maps to mantle viscosity (e.g., Austermann et al., 2013, 2021; Gomez et al., 2018), we compare GIA output from Gom18 to GIA output that utilizes a 1-D reference Earth model (Gomez et al., 2018). Although we are not exploring uncertainty within the 3-D Earth model of Gom18, comparing the 3-D and 1-D GIA output of Gom18 provides context for how incorporating 3-D Earth structure may affect the GIA output of Gol14 and W12.

L119-120: Explain in more detail how LGM ice stream flowlines are defined based on both the reconstructions and the present-day ice flow.

We now specify (L 136-139):

> "From the ice divide to the present-day grounding line, ice flowlines are based on observed ice surface velocities (Rignot et al., 2011), and from the present-day grounding line to the edge of the continental shelf, ice flowlines follows reconstructions of paleo-ice flow reconstructed of Anderson et al., (2014)."

L121: Explain interglacial endmembers.

We now clarify in the text (L 140):

> "LGM and interglacial (i.e. present-day) endmembers."

L130-131: Explain why the understanding of solid Earth-ice sheet interactions would not be possible with traditional ice-sheet modelling methods.

Our ensemble approach with a simple linear stability analysis allows us to assess the stability of all potential grounding line locations across the entire Ross Sea for a range of glacial isostatic adjustment predictions. Traditional ice sheet modeling would require simulating ice sheet evolution over time, which would only predict a single history of grounding line evolution. Given the uncertainty on ice extent over the last deglaciation, our approach allows us to predict a range of GIA-corrected bathymetries, and then assess the stability of possible grounding line locations for all flowlines across the Ross Sea. This approach permits exploration of various scenarios of glacial isostatic adjustment for many possible grounded ice extents.

L136: Why is it spaced at 1 km if the resolution of the ice thickness is lower. How does it improve results to use a 1 km resolution along the flow line?

The 1 km spacing was an error in the text. All high-resolution stability analysis is run at 500 m resolution, and this text has been updated to reflect that (L 155).

> "…to test for stability along each ice stream flowline at 500 m-spaced nodes"

While the resolution of the ice thickness histories is much lower than 500 m (14-20 km), our glacial isostatic adjustment predictions are sensitive to the broader (hundred kilometer scale) loading history, and produce changes in topography with smooth signals that can be interpolated onto higher resolution topography. Indeed, ice sheet loads on a spatial scale smaller than the effective lithospheric thickness will not cause crustal deformation. Therefore, even though our ice history input and our glacial isostatic adjustment model have coarse spatial resolution, it is possible to produce high spatial resolution bathymetric reconstructions using interpolated maps of our glacial isostatic adjustment predictions. We have explained this concept in the main text (L 98-99):

> "The resulting GIA output varies smoothly across spatial scales much broader than the spatial scales of Ross Sea Embayment bathymetric changes."

L136-138: Could you clarify the sample size used for each parameter? Are the values evenly spaced within the given range? Additionally, for clarity, it may be helpful to remove variables from Table 1 that do not have specified values.

We now specify the sample size used for each parameter in Table 1. We choose to keep all parameters within Table 1 to provide the user an easy location to view all parameters, however we do group parameters with specified values together for easier reading.

We also specify our sampling (L 155-159):
> "We consider different combinations of accumulation, basal friction, and ice-shelf buttressing parameters (Table 1) by uniformly sampling the range of basal friction and ice-shelf buttressing and sampling the range of accumulation rates with a non-linear spacing, since they range over an order of magnitude…"

L145: Variable b is not defined.
We now define (L 170-171):
> "b is bathymetric depth at the grounding line"

L149: Insert "and" instead of comma before "ice-shelf buttressing".
Corrected.

L174-176: It is not entirely clear how a zone itself is defined and how the zones are defined as stable or unstable.

In the original text we distinguished between mathematical steady states and how our analysis provides locations at which grounding lines were likely to have persisted or slow down retreat for prolonged time periods. To further make this distinction clear we now use the term "zones of potential persistence" instead of "zones of potential stability".

This is a key concept in our manuscript, and in response to this comment, we have edited the text to provide more detailed descriptions defining our zones of potential persistence. Zones are defined as a 50 km reach along a flowline representing in ice stream transect (Figure 2), or 20 km x 20 km grid cell within the Ross Sea Embayment (Figure 3). A zone is considered mathematically stable if our linear stability analysis produces a "stable steady state" within the zone. A zone is considered a zone of potential persistence based on the number of "stable steady states" it contains, with more "stable steady states" increasing the potential for persistence.

We now provide a more explicit definition of "zone of potential persistence" and explain how we determine whether a zone is considered stable (L 270-273):
> "Zones along the transect with higher counts of potential stable grounding line locations are stable across a wider range of input parameter combinations and therefore have a relatively higher likelihood of persisting at that location regardless of parameter uncertainty and are referred to as "zones of potential persistence""

L179: The parameter space has been explored by ice sheet models as well, for example Albrecht et al. (2020) performed hundreds of simulations for which they systematically varied the parameters using full-factorial parameter sampling. Do you mean here that sampling a wide range of parameter space is not feasible with ice-sheet models on a relatively high resolution? Please clarify.

Yes, indeed. We clarify that we are able to explore a large parameter space while maintaining a relatively high resolution bathymetry, as opposed to Albrecht which utilized a resolution of 16 km (L 59-64):

> "As a result, ensemble runs of ice sheet or coupled GIA-ice sheet models that can encompass uncertainties in climate and glaciologic conditions are run at relatively coarse resolution (16-40 km; Albrecht et al., 2020, 2024; van Calcar et al., 2023; Gomez et al., 2018; Pollard et al., 2017), and are unable to resolve smaller scale bathymetric features, such as pinning points ($\leq$5 km; e.g., McKenzie et al., 2023), which influences grounding line evolution."

Albrecht, T., Winkelmann, R., and Levermann, A.: Glacial-cycle simulations of the Antarctic Ice Sheet with the Parallel Ice Sheet Model (PISM) – Part 2: Parameter ensemble analysis, The Cryosphere, 14, 633–656, https://doi.org/10.5194/tc-14-633-2020, 2020.

L204-206: This is not clear from figure S2, please be more specific on which location this can be seen.

In the Supplementary Material we now specify:

> "The simple grounding line stability model (main text) predicts grounding line stability for Gom18 paleo-bathymetry near the edge of the continental shelf (~1,600 km; Figure S2b) and near the present-day grounding line (~800 km; Figure S2b), and the ice stream flowline model produces the smallest grounding line retreat rates and grounding line discharges at these locations as well. A similar spatial pattern emerges with present-day bathymetry (black; Figure S2). The simple grounding line stability model also predicts a small number of potential stable grounding line zones for present-day bathymetry between 900–1,400 km downstream (Figure S2), but no potential stable grounding line zones over this reach for 20 ka paleo-bathymetries. Over the same reach, the flowline model produces grounding line retreat rates over present-day bathymetry that are ~5-10 times slower than those over 20 ka paleobathymetry."

L300-307: Also discuss how this effect the results of the Gom18 GIA output, since this sea level change does not include the effect of the northern hemisphere.

The Gom18 ice sheet history does include the effects of northern hemisphere ice sheets. Gomez et al. 2018 references Gomez et al., 2013 for methods of applying their sea-level model, which states:

> "In the coupled simulation, changes in the distribution of grounded ice in Antarctica are passed to the sea-level model by the ice-sheet model. However, sea level in the vicinity of the AIS is impacted by changes in ice loading both locally and globally. Thus, in the results presented here, we adopt the ICE5G model (Peltier, 2004) to prescribe the space–time geometry of ice complexes outside of the AIS over the last 40 ky." (Gomez et al., 2013)

L312: Not clear how this is analyzed, since grounding zones were only analyzed at 20 ka and present day.
We clarify how this metric is calculated in the supplementary material:
"We term, "emergent" zones of potential persistence as zones of potential persistence that are stable at present-day, but not stable at a specific previous time in the deglaciation (e.g. zones that are "emergent" at 15 ka are stable at present-day but not at 15 ka). For each time-step we calculate the number of "emergent" zones of potential persistence for that specific timestep. We then normalize all times by dividing by the maximum number of "emergent" zones of potential persistence at a given timestep, which occurs at 12.5 ka for Gol14 and Gom18, and at 10 ka for W12."

L330: Please define "grounding line depth" explicitly.
We define the grounding zone depth as (L 38-39):
"…where ice transitions from grounded to floating"

Figures

Fig. 2: Concerning panel b, please clarify which present day topography is shown, is it observed present day topography? Is the modelled present-day topography by Gomez et al. (2018) equal to the observed present day topography? Please indicate in the text how the difference in present day topography between modelled and observed topography might affect your results.
We now clarify in the figure caption:
"Present-day bathymetry of the Ross Sea Embayment (Morlighem et al., 2020)"

We believe that the ability or inability of the Gomez et al., (2018) model to reproduce observed present-day topography is outside of the scope of this manuscript. Running our persistence analysis over observed present-day bathymetry down sampled to 20 km may produce different spatial patterns than if we used the 20 km modeled present-day bathymetry of Gomez et al., (2018). However, the scope of our manuscript is to explore the use of high-resolution bathymetry and differences that arise with coarser-resolution bathymetry, and so we do not include a discussion on this so as to not distract from the main theme of the manuscript.

Fig. S1: Include in the caption which flow lines are shown and where they are taken from. Also include that the present-day grounding line is shown and explain where it is taken from or how it is computed.
We now clarify in the figure caption
"Flowlines are identical to in main text. Black contour represents present-day grounding line from MEaSUREs (Rignot et al., 2011)."

Fig. S2: Label of the x axis is missing.
Corrected.

---

## Author Comment (AC2)

We thank the Editor and the Referees for their time and their positive and constructive feedback of the manuscript. In response to these reviews, we have edited the text to clarify our arguments and the results of the study. In particular, we have added text to the introduction that contextualizes the motivation of our study for a broader audience. Further, we have addressed all minor revisions suggested by the Referees. We believe that the revised manuscript is improved relative to the original.

In the following, we address each of the comments raised in the reviews and provide a detailed listing of the associated revisions to the text. We intersperse the reviewers' comments (black font) with our responses (blue font).

**Referee #1**
The authors present an analysis of the impacts of (predominantly) GIA on the grounding line stability of Antarctica's Ross Ice Shelf region when combined with high-resolution bathymetry. They do this by combining a range of GIA predictions with a computationally fast (and simplified) ice model. The work shows that specific regions have a greater likelihood of producing stable grounding lines at both 20ka and present-day and these are largely invariant to the exact GIA model employed. They also show that high-resolution (500m) bathymetry produces substantially different regions of grounding line stability than those computed after downsampling the bathymetry to 20km resolution as is typical in coupled ice-GIA models. The paper therefore reaches some important conclusions. The work is well described and the figures are nicely prepared.
We thank the Referee for their positive comments.

While I am not an expert on ice modelling, it appears appropriate and suitable for the task. A range of tests in the supplementary material confirm the results to be robust to various choices.

I have just a few comments where further clarification or discussion is required.

More substantial remarks

L110 - this section, do note that Nield et al 2016 GJI put some constraint on upper mantle viscosity in this region as >10^20 Pa s. That said, their work considered late Holocene flow switching but not the more recent finding of large-scale retreat and readvance, which may affect their conclusions. On that general topic, how does the absence of these large mid to late Holocene signals from the ice models change anything? Perhaps not at all, but maybe worth noting when introducing the ice models.
We now include the Nield et al 2016 reference (L 124-127):
> "Our Earth model is similar to the best fit 1-D Earth model used in Whitehouse et al. (2012), which was determined by inverting for the solid Earth rheology that best fit Antarctic deglacial sea-level data, and is consistent with local constraints on upper mantle viscosity (Nield et al., 2016).."

We do not consider any ice histories characterized by large-scale retreat and re-advance during the Holocene, and it is possible that such loading changes in the Holocene would impact our predictions of grounding line persistence zones at present day. Given the

uncertainty in the timing and magnitude of Holocene readvance, such a consideration could be the subject of future research.

In general, I thought the importance of far-field sea level was emphasised more than the impact of the forebulge collapse (on the shelf break region). I didn't see evidence to suggest one was more important than the other. Please review all mentions or add some extra tests if the distinction matters.

We agree that this is an important point to make in our manuscript. To address this issue, we have added more detail to the text to compare the relative contributions of peripheral bulge collapse and sea level rise caused by global ice sheets melting (L 311-323):

> "There are two GIA mechanisms that cause the locations of zones of potential persistence to shift upstream over the last deglaciation: 1) sea-level fall caused by rebound of the solid Earth under the locus of ice mass loss (~250 m) and 2) sea-level rise caused by global ice sheet melt (~130 m), and secondarily by the collapse of the Antarctica peripheral bulge (~10 m; Figure S3). Within the interior of the Ross Sea Embayment, relative sea-level fall caused by isostatic rebound is nearly twice the magnitude of sea-level rise caused by global ice sheet melt. In contrast, near the continental shelf edge, relative sea-level rise is dominated by global ice sheet melt and peripheral bulge collapse contributes only <10% of the signal.
>
> Isostatic rebound shoals bathymetry within the interior of the Ross Sea Embayment, decreasing flux through the grounding line, thus increasing the potential for a "stable steady-state" grounding line at present-day. Sea-level rise caused by far field ice sheet melt and Antarctic peripheral bulge collapse causes increased grounding line flux, which decreases the likelihood of a "stable steady state" grounding line position near the edge of the continental shelf at present day (Figure 3)."

We now include a new Supplemental Figure (S3) illustrating the magnitude of sea level rise induced by the peripheral bulge.

[Figure]

**Figure S3 | Relative Sea Level (green) decomposed into solid Earth (R; blue) and geoid (G; including direct gravitational effect and global sea-level change; orange) components for a) Gol14 and b) W12 ice histories. a) Map of Antarctica showing location of relative sea-level curves for the interior (I) and peripheral bulge (P). b/d) Relative sea-level curves for W12 ice history. c/e) Relative sea-level curves for Gol14 ice history.**

Minor remarks:

L75 please clarify the method for shifting the bathymetry to 20ka. Do you take model(PD)-model(20ka) and apply that to bedmachine? I guess that is the only approach one could use to adjust bedmachine but please clarify regardless

The Referee's interpretation is correct and we now specify in the main text (L 84-87):

> "To reconstruct Ross Sea Embayment 20 ka bathymetry we modify present-day BedMachine v1.38 bathymetry (500 m horizontal resolution; Morlighem et al., 2020) for the spatiotemporal patterns of GIA caused by the deformational, gravitational, and rotational effects associated with changes in ice load (i.e. $bathymetry_{20\ ka} = bathymetry_{pres} - relative\ sea\ level_{20\ ka}$)."

L85 sedimentation is mentioned along with sea level (and later far field sea level effects) but maybe it is worth adding other things that could contribute to sea level over LGM timescales such as changes in ocean dynamic topography and thermosteric effects.

The Referee raises an important point about other processes that might impact relative sea level in the Ross Sea. However, the other potential processes suggested by the Referee would be of second order (e.g. thermosteric effects would contribute a maximum of ~2.5 m of sea level rise since the Last Glacial Maximum; Simms et al., 2019), so we choose to omit them from the main text.

> Simms, Alexander R., Lorraine Lisiecki, Geoffrey Gebbie, Pippa L. Whitehouse, and Jordan F. Clark. "Balancing the Last Glacial Maximum (LGM) Sea-Level Budget." *Quaternary Science Reviews* 205 (February 1, 2019): 143–53. https://doi.org/10.1016/j.quascirev.2018.12.018.

L114 check the wording of this sentence as I did not understand it

We rewrote the sentence to address this comment, and a comment from Referee #2 (L 128-130):

> "…VM5a Earth model (lithosphere thickness of 96 km, average upper and lower mantle viscosity of 0.5 x $10^{21}$ and 1.6 x $10^{21}$, respectively (Peltier et al., 2015)"

L129 the use of 'geologic record' was confusing to me given the context is present day. That raised the question as to the meaning of 'present day' in the paper more generally. is it within the last few hundred years? Is there a definition you wish to use?

In our study "present day" refers to the time period over which the observations that led to BedMachine were taken over, or the period of time that BedMachine bathymetry is representative of, which is likely the past few years to decades. We agree that "observational record" is a more appropriate term than "geologic record", and now clarify the text to read (L 148-150):

> "…the observational record (e.g. predicting stable steady state grounding lines on present-day bathymetry offshore of the present-day grounding line)"

L150 should flow come before law here in L151?
Corrected.

Equation 5. I looked at the right side of Eq 1 and could not see how taking the derivative with respect to L would arrive at this equation. Please check but given the authorship's mathematical

expertise, it is likely I was missing something. I note that h is defined only in table 1 and not in the text.

Part of the confusion may have been that $h_g$ must first be moved to the right-hand side before taking the derivative. Additionally, $h_g$ is a function of L (since bed elevation and therefore grounding line ice thickness vary downstream). We also now change h to b for clarity (since it references bathymetric depth) and refer to it in the main text. Here is the complete derivation, which we have now included in the Supplementary Text:

$$\frac{d}{dL}\left(PLh_g^{-1} - \Omega h_g^{\beta-1}\right) \tag{1}$$

Applying the chain rule to $PLh_g^{-1}$

$$Ph_g^{-1} - PL\frac{dh_g}{dL}h_g^{-2} \quad -\frac{d}{dL}\left(\Omega h_g^{\beta-1}\right) \tag{2}$$

Applying the chain rule to $\Omega h_g^{\beta-1}$

$$Ph_g^{-1} - PL\frac{dh_g}{dL}h_g^{-2} \quad -(\beta-1)\Omega h_g^{\beta-2}\frac{dh_g}{dL} \tag{3}$$

Applying that $h_g = -\frac{\rho_w}{\rho_i}b$, where b is a function of L

$$Ph_g^{-1} + \left[PLh_g^{-2} + (\beta-1)\Omega h_b^{\beta-2}\right]\frac{\rho_w}{\rho_i}\frac{db}{dL}$$

The original manuscript contained a $\Omega h_b^{\beta-1}$, but now correctly contains $\Omega h_b^{\beta-2}$.

Fig 2. a) Please add some distance markers so one can understand the profile in b). In b) xaxis distance from where?

Thank you for this suggestion. We now include a scale bar and an arrow indicating which flowline is shown in panel b. Distance from the ice divide is shown on the x-axis.

[Figure]

L246 some comment on the forebulge change on the right side of 2b would be appropriate.
We have now added a supplemental figure (Figure S3; see above)
that identifies the magnitude of peripheral bulge collapse, and include a discussion in both the
main text (L 311-323; see above) and the supplementary material (Text S5):

> "On the peripheral bulge RSL is predominantly caused by global sea level rise
> due to global ice sheet melt, which causes ~130 m of sea level rise. Peripheral
> bulge collapse only causes ~10 m of RSL rise, an order of magnitude smaller.
> However, these forcings occur at different times. RSL rise due to changes in
> global sea level occur primarily from 20-10 ka, while changes in RSL due to
> peripheral bulge collapse occur from 8 ka to present-day"

L339 T-test to t-test
Corrected

L358 the methods used to resample are missing. I presume this is some sort of mean. I wonder if
using something other than the mean (like first quartile or max) may produce more realistic
subsampling and useful advice to those running lower-resolution models out of computational
necessity. maybe that messes with ice-ocean melt in those models.
We describe our sampling in the methods section (L 213-215):

> "We resample by smoothing the transect using a windowed mean of the desired resolution and then resampling the smoothed bathymetry at the desired resolution."

The Referee raises an interesting point about more realistic ways of down-sampling or subsampling bathymetry that could be interesting to explore in future work.

Figure 5 The definition of misfit is in the caption but missing from the text. Including it would help the reader avoid confusion
Thank you for this suggestion. We now also include the definition within the main text (L 422-425):

> "Figure 6a shows the percent misfit for change in zones of potential persistence of 20 km resolution compared to 500 m resolution ($\frac{ZPP_{500m} - ZPP_{20km}}{ZPP_{500m}}$)."

L390 add cross reference to Fig 6b
Added.

L389 is 'logarithmic' strictly or is this by eye?
This interpretation was by eye. We now reword to be more specific:

> "We also vary the coarseness of the grid resolution and find that as grid resolution coarsens, percent misfit converges towards ~80-90% (compared to 500 m bed resolution; Fig. 6b)…"

L396 there's already a median misfit of 25% at 1km so is it robust to say 1km? There does not appear to be convergence evident in Fig 6b and so I think you could argue that 500m may not be a high enough resolution. Correct? You could test that with some simulated higher-resolution topography. I guess.
In response to this comment, we have edited the text accordingly. Since "robust" could be interpreted differently, we instead specify the range of misfit and let the reader decide what level is appropriate for their research. While we could test a higher-resolution bathymetry, we feel that limiting our analysis to 500 m is appropriate given the resolution of BedMachine and realistic computational constraints. Assessing the effects of finer scale topography could be interesting, but as the reviewer notes this would have to be synthetic topography, which would in turn require observations or assumptions about the roughness of the true bathymetry and is beyond the scope of our study (L 439-441).

> "Based on our analysis, compared to 500 m grid resolution, a grid resolution of 20 km leads to a ~95% misfit, while a grid resolution of 1 km leads to a ~25% misfit (Fig. 6b)."

L415 'offshore Victoria Land' could be Ross Sea or Southern Ocean. please be more specific
Thank you for pointing out this confusion. We have now removed this reference to location so that the text reads (L 459-460):

> "The grounding line then retreated throughout the Holocene (Bart et al., 2018; Halberstadt et al., 2016; Prothro et al., 2020)."

L444-447 I found this sentence unclear as to its meaning.
We rephrase to clarify (L 486-490):

"In this study we have shown that, in addition to impacting transient grounding line retreat, glacial isostatic adjustment promotes stability across ice-age timescales by shifting zones of potential grounding line persistence from near the edge of the continental shelf toward the present-day grounding line across the deglaciation."

**Supp Material**
S2 'marine ice sheet evolution' does not make sense to me
We now clarify:
> "we run a 1-D marine ice sheet, shallow-shelf approximation, flowline model"

Supp Table I think should be labelled Table S1 rather than STable 1.
Corrected.

Relevant here and in the main paper, where do these values come from and does it matter if they are not realistic? I think so. I presume for instance that the SMB present-day is actually about 0.1m/yr (ice? water?)
We now cite references to justify out choice of SMB values in the main text (L 170-171):
> "averaged surface mass balance 0.01–0.3 m/yr (Buizert et al., 2015; Cavitte et al., 2018)"

Present-day SMB within the Ross Sea Embayment is ~0.1 m/yr, however, rates can be as high as 0.3 m/yr (Lenaerts et al., 2012). To isolate the role of GIA, we consider the full range of SMB inferred from ice core and radar records for the time period spanning the LGM to present-day (Buizert et al., 2015; Cavitte et al., 2018).

> Buizert, C., K. M. Cuffey, J. P. Severinghaus, D. Baggenstos, T. J. Fudge, E. J. Steig, B. R. Markle, et al. "The WAIS Divide Deep Ice Core WD2014 Chronology – Part 1: Methane Synchronization (68–31 Ka BP) and the Gas Age–Ice Age Difference." *Climate of the Past* 11, no. 2 (February 5, 2015): 153–73. https://doi.org/10.5194/cp-11-153-2015.

> Cavitte, Marie G. P., Frédéric Parrenin, Catherine Ritz, Duncan A. Young, Brice Van Liefferinge, Donald D. Blankenship, Massimo Frezzotti, and Jason L. Roberts. "Accumulation Patterns around Dome C, East Antarctica, in the Last 73 Kyr." *The Cryosphere* 12, no. 4 (April 17, 2018): 1401–14. https://doi.org/10.5194/tc-12-1401-2018.

> Lenaerts, J. T., Van den Broeke, M. R., Van de Berg, W. J., Van Meijgaard, E., & Kuipers Munneke, P. (2012). A new, high-resolution surface mass balance map of Antarctica (1979–2010) based on regional atmospheric climate modeling. *Geophysical research letters*, *39*(4).

Fig S4 define ZPS on yaxis label
Corrected.

---

## Author Comment (AC3)

We thank the Editor and the Referees for their time and their positive and constructive feedback of the manuscript. In response to these reviews, we have edited the text to clarify our arguments and the results of the study. In particular, we have added text to the introduction that contextualizes the motivation of our study for a broader audience. Further, we have addressed all minor revisions suggested by the Referees. We believe that the revised manuscript is improved relative to the original.

In the following, we address each of the comments raised in the reviews and provide a detailed listing of the associated revisions to the text. We intersperse the reviewers' comments (black font) with our responses (blue font).

**Referee #3**
I apologise for the late posting of this review.

This manuscript presents a series of idealised grounding line stability simulations for the Ross Sea Embayment, both with contemporary and palaeo bathymetry reconstructions. The authors show that isostatic depression at the LGM restricts the potential areas of grounding line stability, and that the areas allowing stability increase at the present day due to local relative sea level fall since the LGM. Later results demonstrate the importance of horizontal resolution to this problem; coarser (20 km) resolution underestimates the frequency of potential areas of grounding line stability.

Arguably, each individual element of the results is not surprising. But the work is comprehensive in its exploration of the topic. The exploration of the resolution dependency, for example, is robust and detailed. Simplifications in the grounding line flux simulations are well justified, and the experimental design allows for clear comparisons to be drawn. These technical choices are clearly described in the text.

We thank the Referee for these positive comments.

The introduction might benefit from a more detailed introduction to GIA, and there are some sections where added clarify would be useful (highlighted below). As such, I recommend publication after minor revisions.

Main points

As someone whose knowledge of GIA comes through ice sheet modelling (rather than my first-order research focus), it might be beneficial to provide some more detail and background in the introduction on GIA and Antarctica, including a brief review of the recent advances in coupled modelling (at the moment this is limited to a sentence around line 51). The introduction is currently brief enough for you to safely add more background without bloating the section, and it might increase the potential audience of the manuscript and/or increase the usefulness of the manuscript to an audience like me.

In response to this comment and a similar comment by Referee #2, we have added text to the introduction that contextualizes our research study (L 58-70):

"Although coupled GIA-ice sheet models provide valuable insight into ice sheet-solid Earth interactions across Antarctica, they are computationally

expensive. As a result, ensemble runs of ice sheet or coupled GIA-ice sheet models that can encompass uncertainties in climate and glaciologic conditions are run at relatively coarse resolution (16-40 km; Albrecht et al., 2020, 2024; van Calcar et al., 2023; Gomez et al., 2018; Pollard et al., 2017), and are unable to resolve smaller scale bathymetric features, such as pinning points (≤5 km; e.g., McKenzie et al., 2023), which control grounding line evolution. Recent coupled GIA-ice sheet modeling showed that increasing bathymetric resolution (from 2 km to 1 km) slowed predicted grounding line retreat by up to ~20% in the Amundsen Sea; however such high-resolution modeling is more appropriate for smaller regions (the Ross Sea Embayment is ~4x their model domain size) and for timescales of centuries rather than millennia (Houriez et al., 2025). Exploring the impact of high-resolution bathymetry on grounding evolution since the LGM across a larger area, such as the Ross Sea Embayment, is still a computational challenge."

**Specific points**

58: "large- and small-scale bathymetric features", what do you mean by this?
We no longer reference "large-scale" and specify "small-scale" (L 436):
   "…small-scale (≤5 km) bathymetric pinning points"

74: Just BedMachine minus GIA signal? Are there any complications from mismatched grids?
The Referee is correct that we subtract the GIA signal from BedMachine (L 84-87):
   "To reconstruct Ross Sea Embayment 20 ka paleobathymetry we modify present-day BedMachine v1.38 bathymetry (500 m horizontal resolution; Morlighem et al., 2020) for the spatiotemporal patterns of GIA caused by the deformational, gravitational, and rotational effects associated with changes in ice load (i.e. $\text{topography}_{20\ ka} = \text{topography}_{pres} - \text{relative sea level}_{20\ ka}$)."

There are no complications from mismatched grids because the GIA output varies over long wavelengths (~100 km scale) so it can be interpolated to higher resolution without introducing artifacts (L 98-99).
   "The resulting GIA output varies smoothly across spatial scales much broader than the spatial scales of Ross Sea Embayment bathymetric changes."

102: Can you provide some more detail of the Gom18 reconstruction here? An ensemble again? If so, how did you select which members to use/average?
The Gom18 ice sheet model is the result of a single run of the coupled GIA-ice sheet model. We have added additional text about each ice history adopted in our study, including Gom18 (L 113-117):
   "The third ice history Gomez et al. (2018; henceforth Gom18) has a deglacial Antarctic Ice Sheet volume change of ~6 m GMSLE. The Gom18 model is a single model run of a coupled, gravitationally consistent GIA-dynamic ice sheet model that incorporates 3-D earth structure and was forced by climate via benthic $\delta^{18}O$ records. The ice sheet model was run with 20 km resolution and 200 year timesteps."

119: Some more information about how these flowlines are produced would be useful.

We have added text in response to this comment and a similar comment made by Referee #2 (L 136-139):

> "From the ice divide to the present-day grounding line, ice flowlines are based on observed ice surface velocities (Rignot et al., 2011), and from the present-day grounding line to the edge of the continental shelf, ice flowlines follows reconstructions of paleo-ice flow reconstructed of Anderson et al., (2014)."

123: Is the flow reorganization during the deglaciation significant? The justification of keeping the flowlines consistent to aid comparison seems sound, but it would be useful to consider what magnitude of reorganization we're dealing with.

The changes in flow associated with the reorganization was limited to near the present-day outlets of the Transantarctic Mountains (near Drygalski Trough). The evidence shows slight changes in flow direction (Lee et al., 2017), and a general readvance (Greenwood et al., 2018; Lee et al., 2017). These changes are of second order compared to changes in grounding line position associated with the deglaciation, and therefore we would expect would have minimal impact on our findings given their relatively small magnitude. The flow reorganization during the deglaciation would be potentially interesting to explore in future research.

138: How do you statistically sample over the parameter space?

We now specify the sample size used for each parameter in Table 1 and include text describing how we sample the parameter space (L 155-159):

> "We consider different combinations of accumulation, basal friction, and ice-shelf buttressing parameters (Table 1) by uniformly sampling the range of basal friction and ice-shelf buttressing and sampling the range of accumulation rates with a non-linear spacing, since they range over an order of magnitude…"

150: "Nye-Glenn law" > "Nye-Grenn flow law"?

Corrected

339: "T-test" > "t-test"?

Corrected

174-183: I'm not sure how the zones along the flowline were defined?

Zones are defined as a 50 km reach along a flowline representing in ice stream transect (Figure 2), or 20 km x 20 km grid cell within the Ross Sea Embayment (Figure 3). We have added text to make this definition clearer in response to this comment and a similar comment made by Referee 2. We have added the following text (L 270-273):

> "Zones along the transect with higher counts of potential stable grounding line locations are stable across a wider range of input parameter combinations and therefore have a relatively higher likelihood of being stable regardless of parameter uncertainty and are referred to as "zones of potential persistence"

255: "Changes in the density of potential zones of ground line stability" took some time for me to fully understand as a concept. Room to explain it in more detail elsewhere?

In response to this comment, we have rephrased our wording to focus on counts of potential grounding line locations, rather than density. We hope this wording makes the concept more intuitive for the reader.

We have edited the text to clarify (L 262-277):

> "For example, Figure 2b shows the count of potential stable steady-state grounding lines associated with the reconstructed bathymetric transect for the paleo-Whillans ice stream (Fig. 2a) corrected for GIA (Gom18–orange, W12–purple, Gol14–red; Figure 2b), compared to the present-day bathymetry (black; Figure 2b). Present-day bathymetry contains a higher count of potential stable steady states upstream of the flowline, close to the present-day grounding line (black; Figure 2B), compared to any of the 20 ka bathymetries corrected for GIA (orange, purple, and red; Figure 2b), which contain higher counts of potential stable steady states near the edge of the continental shelf.
>
> Zones with higher counts of potential stable steady-state grounding line locations are stable across a wider range of input parameter combinations and therefore have a higher likelihood of being stable regardless of parameter uncertainty. Henceforth, we refer to these locations as "zones of potential persistence" (ZPP). Each zone contains the count of potential stable grounding lines within a 50 km reach along a given ice stream flowline (Figure 2b). The present-day bathymetry has zones of potential persistence near the present-day grounding line (~750 km downstream; black; Figure 2b), whereas each 20-ka bathymetry has zones of potential persistence near the continental shelf break (~1600 km downstream; orange, purple, and red; Figure 2b)."

And we also change the Figure 2 caption wording for clarity (L 286):

> "Zones of potential persistence (ZPP) along individual flowlines".

268: "less" > "fewer"?
Corrected

292: "Our analysis shows…" onwards; this sentence is hard to follow.
We rephrase to (L 324-330):

> During the LGM the grounding line was located near the edge of the continental shelf, and at present-day the grounding line is located within the Ross Sea Embayment interior. Our analysis shows that GIA promotes grounding line stability near the edge of the continental shelf during the LGM and promotes grounding line stability within the Ross Sea Embayment interior at present-day. The co-occurrence of inferred locations of the grounding line for present-day and LGM suggests GIA stabilizes the grounding line at both glacial maximum and interglacial climate states.

Figure 3: Struggling a little bit with the telling apart the "stable for present-day only" and the lighter blues. Could you make it more obviously different?
We have changed the color to purple to make this more obvious visually.

355: 20-40 km (40 is quite common for palaeo), but your later analysis (figure 6, for example) caps out at 20 km?
We now include 40 km resolution in our analysis in Figure 6b.

377: In this section also mention the resolution dependency of simulating GL dynamics, as in MISMIP experiments?

We instead choose to highlight a recent study that show resolution dependency for a coupled GIA-ice sheet model (Houriez et al., 2025) 9L 64-66):

> "Recent coupled GIA-ice sheet modeling showed that increasing bathymetric resolution (from 2 km to 1 km) slowed predictions of grounding line retreat by up to ~20% in the Amundsen Sea (Houriez et al., 2025)…"

Houriez, Luc, Eric Larour, Lambert Caron, Nicole-Jeanne Schlegel, Surendra Adhikari, Erik Ivins, Tyler Pelle, Hélène Seroussi, Eric Darve, and Martin Fischer. "Capturing Solid Earth and Ice Sheet Interactions: Insights from Reinforced Ridges in Thwaites Glacier," January 24, 2025. https://doi.org/10.5194/egusphere-2024-4136.

---

## Author Response (AR1)

We thank the editor for their helpful comments and facilitation of this process and we thank the reviewers for their feedback which improved our manuscript.